# Changes in diet, exercise and psychology of the quarantined population during the COVID-19 outbreak in Shanghai

Li Qiu[1,2&], Chenchen Li[1&], Wen He[1,2], Xuelian Yin[1,2], Lin Zhan[1], Junfeng Zhang[1]*, Yanli Wang [1,2,3]*

**1** School of Environmental and Chemical Engineering, Shanghai University, Shanghai, P. R. China, **2** School of Medicine, Shanghai University, Shanghai, P. R. China, **3** School of Pharmacy & The First Affiliated Hospital, Hainan Medical University, Haikou, Hainan, P. R. China

& These authors contributed equally to this work.
* zjf18722830@shu.edu.cn (JZ); wangyanli@staff.shu.edu.cn (YW)

**Data Availability Statement:** The data relevant to this study are available from Zenodo at DOI: 10. 5281/zenodo.7359688 (https://doi.org/10.5281/ zenodo.7359688).

## Abstract

### Background

In March 2022, a severe outbreak of COVID-19 broke out in Shanghai, with the virus spreading rapidly. In the most severe two months, more than 50,000 people were diagnosed with COVID-19. For this reason, Shanghai adopted three-district hierarchical management, requiring corresponding people to stay at home to contain the spread of the virus. Due to the requirements of prevention and control management, the diet, exercise and mental health of the corresponding population are affected to a certain extent.

### Objectives

This article aimed to understand the population in the diet, exercise and psychological changes.

### Methods

This study carried out the research by distributing the electronic questionnaire and carried out the statistical analysis.

### Results

People reduced the intake of vegetables and fruits (P = 0.000<0.05), people did about an hour less exercise per week on average (P = 0.000<0.05), the number of steps they took per day decreased by nearly 2000 steps (P = 0.012<0.05), and there were significant changes in the way they exercised.

**Funding:** This study was supported by the National Natural Science Foundation of China (81922037, 11575107, 21371115, and 22003038), the Shanghai University-Universal Medical Imaging Diagnostic Research Foundation (19H00100), and Shanghai Biomedical Science and Technology Support Project (19441903600). The funders had no role in study design, data collection and analysis, decision to publish, or preparation of the manuscript.

## Conclusion

In terms of psychological state, people have some depression, anxiety and easy to feel tired after lockdown. This study can also provide reference for policy adjustment and formulation of normalized epidemic management in the future.

## 1 Introduction

In December 2019, a novel coronavirus infection outbreak spread in Wuhan, and in March 2020, the World Health Organization declared COVID-19 a "global pandemic" [1]. In the face of COVID-19, we have always adopted proactive and effective policies to protect people's health. The novel coronavirus spread in Shanghai in March after an imported case contaminated the environment with the virus. In order to quickly and effectively contain the development of the epidemic, Shanghai has taken some measures to reduce the corresponding personnel flow. These measures include lockdown management and home quarantine [2–4]. Shanghai residents have been asked to reduce unnecessary outdoor activities. Most of them stay at home or work from home, which has a certain impact on people's daily life.

In previous studies, it can be known that the lifestyle of the quarantined population will change during the containment period [5–7]; One study found that people generally increased their snacking during quarantine [8]; There was also a marked increase in the consumption of wheat products and salted products [9]; However, the frequency of rice, meat, poultry, fresh vegetables, fresh fruits, soy products and dairy products decreased significantly [10]. Their stress will increase significantly during the containment period [11]. Changes in lifestyle and eating habits as a result of containment can lead to weight gain [12]. Other studies have shown that people are significantly less physically active than they were before the pandemic, as well as overeating [13–15]. There was a significant decrease in physical activity levels [16, 17]. Public health interventions like quarantine and isolation may have impacted mental health during COVID-19 [18]. A study in Vietnam found significantly higher levels of depression, anxiety and stress among people in lockdown due to being single, separated or widowed, higher levels of education, larger family size, unemployment, and contact with potential COVID-19 patients [19]. A study from France says the lockdown affects people's mood, with hope and anxiety being the two ways to deal with uncertainty [20]. The current literature suggests that people affected by COVID-19 may have a high burden of mental health problems, including depression, anxiety disorders, stress, panic attack, irrational anger, impulsivity, sleep disorders and emotional disturbance [21].

From the beginning of March to now, universities and some communities have been under lockdown for months [22–25]. During this period, we found some changes in the life of the quarantined people through social media [26, 27]. In view of the long period of lockdown and the large number of people under lockdown, this paper will study the changes in diet exercise and psychology of people under lockdown management and isolation in Shanghai during the epidemic, and mainly adopt the method of issuing electronic questionnaires [28, 29].

The study aimed to understand the changes in people's diet, exercise and psychology caused by quarantine. Our findings would provide empirical evidence for the following interventions and policies. They may also serve as a reference for other countries to raise awareness of or developing solutions to this issue.

## 2 Materials and methods

### 2.1 Participants

This online survey was distributed by the author in the form of electronic questionnaire through WeChat, a social media platform, in March 2022. Participants were asked to complete electronic questionnaires. The respondents of this survey are limited to residents and teachers and students of colleges and universities in Shanghai under lockdown management. The research protocol was approved by the ethics committee of Shanghai University. All participants provide informed consent prior to taking part in this study.

### 2.2 Measures

The questionnaire is divided into four parts. The first part is the basic information of the respondents, including gender, age, height and weight before and after containment. The second part is about the diet and nutritional changes of the respondents before and after the containment. Relevant questions are designed with reference to the Dietary Guidelines for Chinese Residents, including the types of food and the amount of food to measure the diet and intake status of the respondents. The third part is the movement changes of the respondents before and after the containment control, from the exercise mode, exercise duration, exercise intensity and other dimensions, to measure the movement changes of the respondents before and after the containment control; The fourth part is the change of the psychological state of the respondents before and after the containment. The design is based on the PHQ-9 depression screening scale to evaluate the psychological state of the respondents. The second, third and fourth parts of the questionnaire were answered using the Richter scale, ranging from very unsatisfactory 1 to very satisfactory 5. Some questions were divided into 1, 2 and 3 points. The main questions of the survey are shown in Table 1.

**Table 1. The main questions of the survey.**

| |
|---|
| Q: How many kinds of food do you eat every day? |
| A: 1.0–2 species 2.3–5 species 3.6–8 species 4.9–11 species 5.12 species or more |
| Q: How much vegetables do you eat every day? |
| A: 1. 0–50 grams 2.51–100 grams 3.101–200 grams 4.201–299 grams 5.300 grams or more |
| Q: How much fruit do you eat every day? |
| A: 1. 0–50 grams 2. 51–100 grams 3. 101–150 grams 4. 151–199 grams 5. 200 grams or more |
| Q: How many hours do you exercise every week? |
| A: 1. Less than 1 hour 2. 1–2 hours 3. 2–3 hours 4. 3–4 hours 5. More than 4 hours |
| Q: What are your main sports activities? |
| A: 1. Track and field walking 2. Ball 3. Swimming 4. Fitness equipment and gym 5. Other |
| Q: How many steps do you take every day? |
| A: 1.0–1500 steps 2.1501–3000 steps 3.3001–4500 steps 4.4501–5999 steps 5.6000 steps and above |
| Q: Little interest or pleasure in doing things? |
| A: 1. Strongly disagree 2. Disagree 3. Neither agree nor disagree 4. Agree 5. Strongly agree |
| Q: Feeling down, depressed, or hopeless? |
| A: 1. Strongly disagree 2. Disagree 3. Neither agree nor disagree 4. Agree 5. Strongly agree |
| Q: Trouble falling or staying asleep, or sleeping too much? |
| A: 1. Strongly disagree 2. Disagree 3. Neither agree nor disagree 4. Agree 5. Strongly agree |
| Q: Feeling tired or having little energy? |
| A: 1. Strongly disagree 2. Disagree 3. Neither agree nor disagree 4. Agree 5. Strongly agree |

## 2.3 Data analysis

Statistical data were analyzed using SPSS. Before statistical analysis of the results, the reliability of the questionnaire results was firstly analyzed. The Cronbach's Alpha value obtained was 0.781, and the Cronbach's Alpha value based on the standardized item was 0.782, both greater than 0.75, indicating good reliability of the scale and good stability and reliability of the questionnaire. Descriptive statistical analysis and analysis of variance were mainly used for data analysis. Independent sample T-test was used for comparison of independent sample means, and P value < 0.05 was considered statistically significant.

# 3 Results

## 3.1 Sample characteristics

A total of 625 valid questionnaires were collected for this survey (invalid questionnaires have been removed). The trend of COVID-19 in Shanghai and the information of respondents are shown in Fig 1. Among them, 283 questionnaires were from males (45.28%) and 342 from females (54.72%). Among all the respondents, 414 people aged 18–29 occupy the majority, accounting for 66.24% of the total, followed by 78 people aged 30–39, accounting for 12.48%, 52 people aged 40–49, and 42 people aged 50–59. There were 27 under 18 and 12 aged 60 and over respectively.

## 3.2 Changes in diet

Before and after lockdown, there were significant changes in people's diet and nutrition intake. Table 2 shows the changes in people's diet before and after lockdown. It can be seen that people's food mix after lockdown was not as diverse as before (P = 0.000<0.05, N = 625), and the types of food they consumed per day were also reduced compared with before lockdown (P = 0.002<0.05, N = 625), and people generally ate less after lockdown than before (P = 0.001<0.05, N = 625), and the taste was lighter, which was reflected in the decrease in the daily intake of salt and sugar (P<0.05, N = 625), and in the intake of fruits and vegetables, there was a trend of decrease after the control (P<0.05, N = 625).

## 3.3 Changes in physical activity

Once the lockdown measures are implemented, people will only be able to move around the house they live in, so sports and fitness activities will be limited. In the analysis of the questionnaire results (see Table 3), we found that people's weekly exercise time decreased significantly (answer option 1 = less than 1 hour, 2 = 1–2 hours, 3 = 2–3 hours, 4 = 3–4 hours, 5 = more than 4 hours), and the weekly exercise time decreased 1–2 hours after containment, as shown in Fig 2 (P<0.05, N = 625). Similarly, the number of walking steps per day also decreased significantly (1 = 0–1500 steps, 2 = 1501–3000 steps, 3 = 3001–4500 steps, 4 = 4501–6000 steps, 5 = more than 6000 steps). Due to space limitation, the number of walking steps per day decreased by more than 2000 steps (P<0.05, N = 625). Secondly, we also investigated the changes of people's exercise styles, as shown in Fig 3. Before the control management, people's exercise and fitness styles were diversified and balanced, including track and field walking, ball games, swimming, gym and equipment sports, etc. However, after the lockdown management, people can hardly carry out swimming exercise; Track and walking decreased by 2 percent and ball games by 4 percent; Three percent increased their use of fitness equipment; Another 15 percent added other forms of exercise, such as calisthenics, that can be done indoors without equipment.

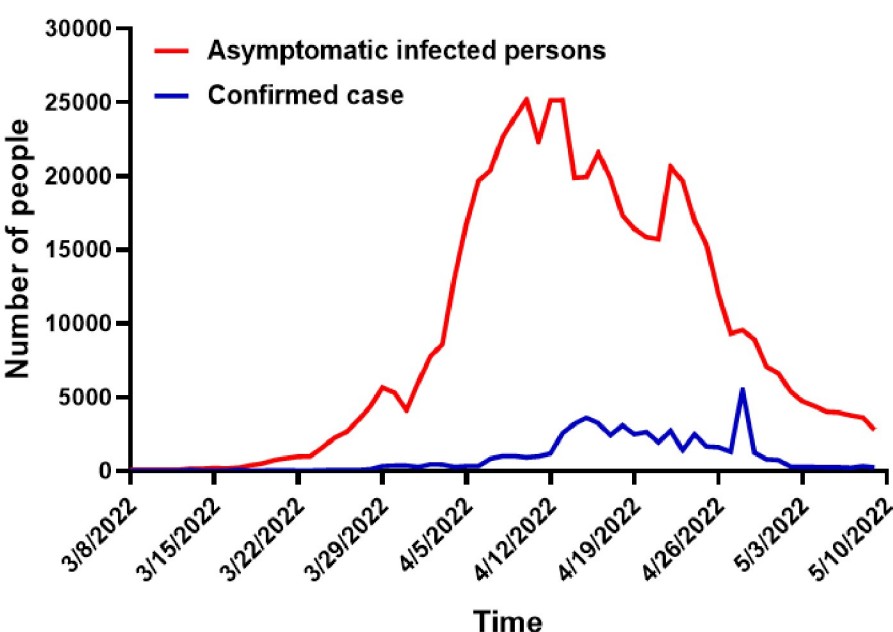

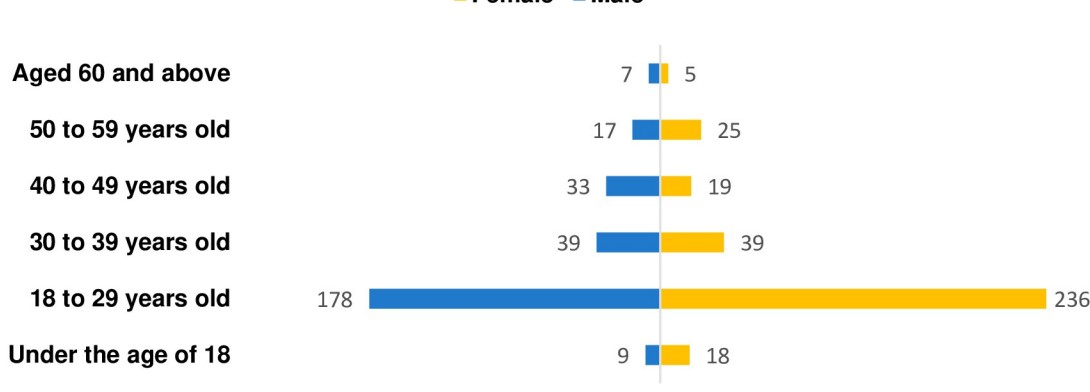

**Fig 1. Trend of COVID-19 in Shanghai (a) and information of the study participants (b).**

### 3.4 Changes in mental state

In terms of changes in psychological state, there were also significant changes before and after containment, as shown in Table 4. After containment, people were more likely than before to feel depressed, depressed or hopeless (P<0.05, N = 625), and more likely to feel tired or inactive (P = 0.002<0.05, N = 625). The results showed that people were more prone to emotional fluctuations after containment (P<0.05, N = 625), more difficult to control their unpleasant emotions (P = 0.008<0.05, N = 625), and more likely to feel passive and unmotivated after containment (P = 0.001<0.05, N = 625). In addition, another interesting finding is that when we calculate the total score of psychological part of respondents before containment in the

Table 2. Differences in diet before and after closed management.

| | Before the closed management | | | After the closed management | | | p-value | T-test Sig. (two-sided test) |
|---|---|---|---|---|---|---|---|---|
| | mean | min | max | mean | min | max | | |
| Balanced match of meat and vegetables, intake of various kinds of foods | 3.90 | 1 | 5 | 3.67 | 1 | 5 | 0.000 | 0.000 |
| The types of food you eat each day | 3.50 | 1 | 5 | 3.24 | 1 | 5 | 0.002 | 0.000 |
| You usually eat how full | 3.78 | 1 | 5 | 3.64 | 1 | 5 | 0.001 | 0.019 |
| Your food tastes light | 3.50 | 1 | 5 | 3.64 | 1 | 5 | 0.023 | 0.038 |
| You eat salt every day | 1.39 | 1 | 2 | 1.36 | 1 | 2 | 0.037 | 0.000 |
| You eat sugar every day | 1.83 | 1 | 3 | 1.79 | 1 | 3 | 0.000 | 0.000 |
| You often eat fruits and vegetables | 3.84 | 1 | 5 | 3.45 | 1 | 5 | 0.000 | 0.000 |
| You eat vegetables every day | 3.23 | 1 | 5 | 2.94 | 1 | 5 | 0.000 | 0.000 |

questionnaire, those with a score of less than 25 are regarded as people with a good psychological state, and those with a score of more than 25 are considered to be anxious (Fig 4). We found that people with good psychological state before containment would have some anxiety, depression and difficulty in sleeping after containment (P<0.05, N = 625). However, those who were anxious before were less anxious and depressed after containment (P<0.05, N = 625).

## 4 Discussion

Since the spread of COVID-19 in late 2019, the epidemic has been managed on a regular basis to ensure people's normal work and life. Under this management mode, we always adhere to the dynamic zero clearance policy in the face of sudden local epidemic. Among the novel coronavirus strains in Shanghai, the omicron variant is highly infectious, stealth and rapid in transmission, making it difficult to detect at once and thus causing social transmission. Under such circumstances, it is very necessary to prevent the rapid spread of the virus in a timely manner. Therefore, appropriate containment measures should be implemented for communities and units at risk of virus transmission. In previous studies, researchers have reported that semilockdown increases body fat [12]; Other studies have shown that although the frequency of exercise and going out in Japan has not changed significantly during the epidemic, the epidemic has brought more stress to people [4]. The results of a survey from Israel showed that while depression increased during the COVID-19 pandemic, there was also a significant increase in people's levels of hope [30]. Other studies have shown that lockdown management caused by the epidemic will worsen the sleep quality of people and increase people's negative emotions [30–32]. On the basis of previous studies, our study focused on the changes in diet, exercise and mental status of the corresponding population during the closed management period. Through investigation and analysis, in terms of diet, people's diet and nutrition intake have been changed by containment management [33]. For example, the daily food collocation is not as diverse as before, the food intake is less than before, the taste is generally lighter, and

Table 3. Differences in exercise before and after closed management.

| | Before the closed management | | | After the closed management | | | p-value | T-test Sig. (two-sided test) |
|---|---|---|---|---|---|---|---|---|
| | mean | min | max | mean | min | max | | |
| Total exercise time per week | 3.19 | 1 | 5 | 2.27 | 1 | 5 | 0.000 | 0.000 |
| The number of steps taken each day | 3.60 | 1 | 5 | 2.25 | 1 | 5 | 0.012 | 0.000 |

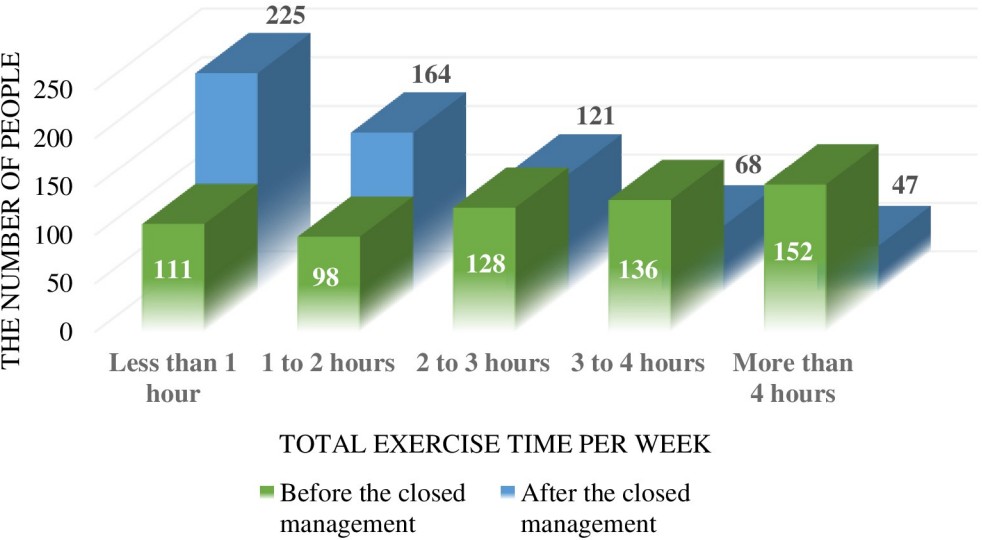

**Fig 2. Total exercise time per week before and after closed management.**

the intake of fruits and vegetables is also less than before. Because people's exercise and fitness can only be carried out at home or in the dormitory [34, 35], people's exercise duration and amount of exercise have been significantly reduced, and their exercise methods have been significantly changed, and outdoor exercise has been significantly reduced. The sudden lockdown also brings a certain change of psychological state to everyone [36–38]. People are more likely to feel depressed, depressed or desperate, tired or not energetic, with greater mood fluctuations [39, 40], more prone to ups and downs, more difficult to control their unpleasant emotions [41, 42], and more likely to feel passive and unmotivated [43, 44]. In addition, we found that people with good psychological state before containment would have some anxiety, low mood, difficulty in falling asleep and other situations after containment [45]; However, people who were previously anxious experienced less anxiety and depression after containment than before.

In view of the above studies, we found that lockdown management due to COVID-19 outbreak would bring certain negative effects on the confined population, resulting in unhealthy changes in diet, exercise and psychology. Therefore, in the following normalized epidemic management, policy makers should take into account the adverse impact of lockdown management on residents, so as to formulate epidemic prevention and control measures that are more conducive to the normal life of the quarantined population, for example: communities can provide services such as delivering fresh vegetables and fruits to people under lockdown. For the closed management of the population can be implemented by time partition measures to provide people with outdoor exercise opportunities; During the lockdown period, psychological counseling should be strengthened for relevant personnel, and psychological hotlines should be kept unblocked or online psychological counseling channels should be opened.

The strength of this study is that it discusses the changes in diet, exercise and psychology brought about by closed management. However, this study also has some limitations, such as a short research period, and due to closed management measures, the questionnaire can only be distributed online, so some people who do not often use social media platforms are not included in the survey. In addition, this study mainly explored the changes in diet, exercise and psychological state before and after lockdown, but did not involve the causes of these

### Exercise mode before closed management

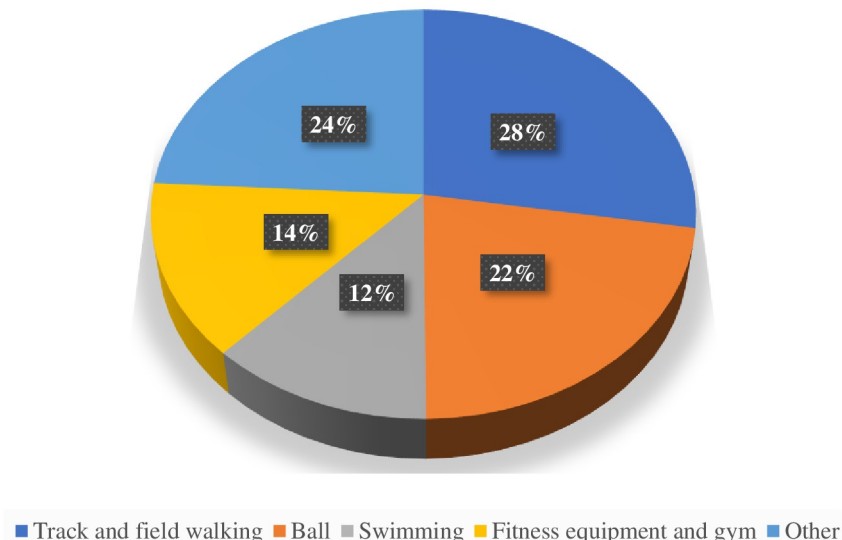

### Exercise mode after closed management

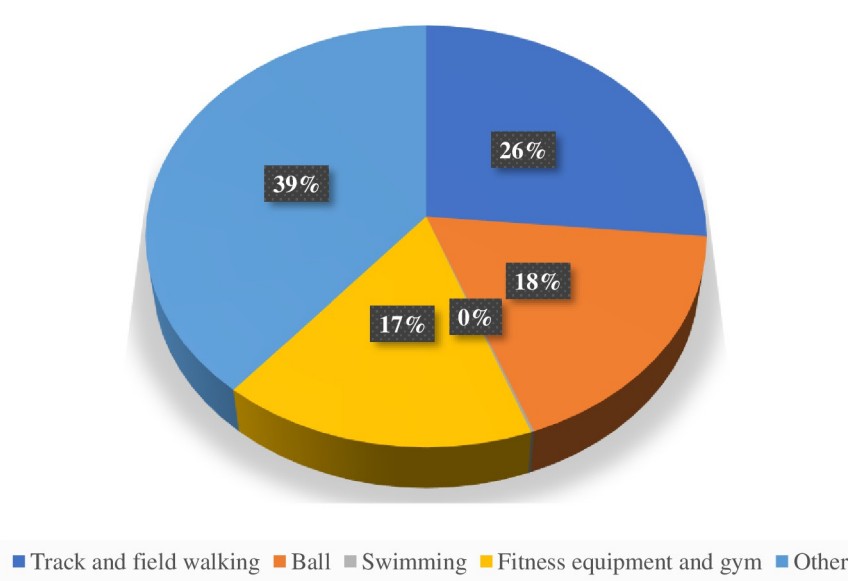

**Fig 3. Exercise mode before and after closed management.**

changes. The influence of closed management on people is not only in the three aspects of diet, exercise and psychological state, but also in the aspects of people's work, study and happiness. In the part of investigating the psychological state of the population under lockdown, this study did not involve the investigation of burnout in the population. Burnout is an important

**Table 4. Psychological changes before and after closed management.**

| | Before the closed management | | | After the closed management | | | p-value | T-test Sig. (two-sided test) |
|---|---|---|---|---|---|---|---|---|
| | mean | min | max | mean | min | max | | |
| Often feeling down, depressed or hopeless | 2.78 | 1 | 5 | 2.97 | 1 | 5 | 0.000 | 0.006 |
| Often feel tired or low in energy | 2.73 | 1 | 5 | 3.10 | 1 | 5 | 0.002 | 0.000 |
| Mood swings are high and easy to rise and fall | 2.79 | 1 | 5 | 2.90 | 1 | 5 | 0.000 | 0.004 |
| It's hard to control your unhappiness | 2.77 | 1 | 5 | 2.82 | 1 | 5 | 0.008 | 0.000 |
| Always feeling passive and unmotivated | 2.70 | 1 | 5 | 2.84 | 1 | 5 | 0.001 | 0.049 |

public health problem, and it is very important to study the burnout of the general public when entering the post-epidemic era [46]. These will be questions that researchers can continue to explore in the future.

## 5 Conclusions

This study mainly discusses whether the containment management brought about by the current outbreak in Shanghai will change the diet, exercise and psychological status of the population under containment. Through our survey, we found that under the control, people had significant changes in the types of food they consumed, the amount of food they ate, the taste, and the consumption of fruits and vegetables was also less than before. Due to the limitation of space, people will spend less time exercising after lockdown, and there will be significant changes in the way of exercise. There is also psychological state, people generally have some low mood, mood swings, passive and so on. Therefore, during the containment period, relevant departments should pay more attention to people's food supply, exercise and psychological state. When the epidemic is stable, they can adjust the containment measures appropriately, so that people can purchase supplies, exercise and adjust their mood, which will also help to normalize the management of COVID-19.

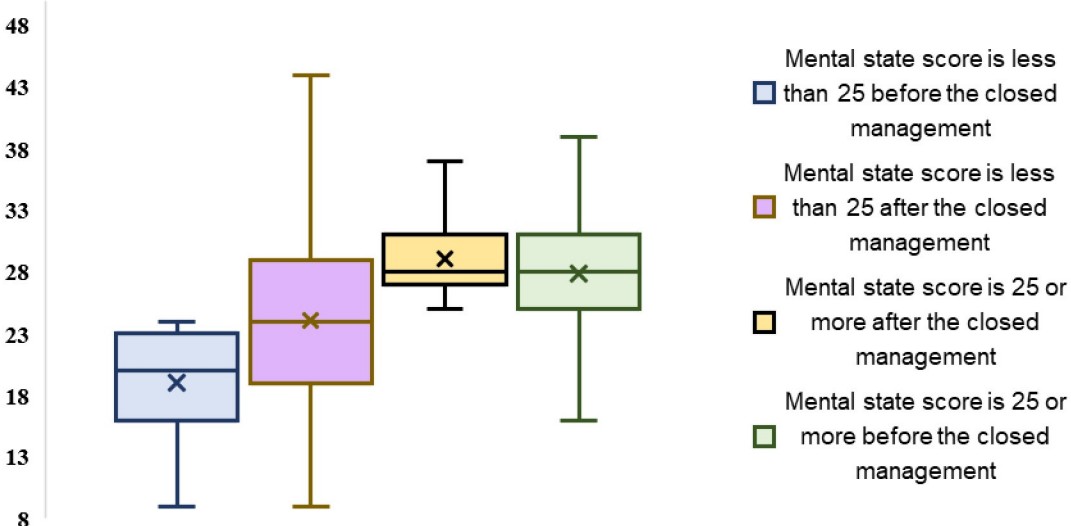

**Fig 4. Psychological changes before and after closed management (People with different mental state before closed management).**

## Acknowledgments

We are very grateful to all participants who volunteered to take part in this study.

## Author Contributions

**Data curation:** Li Qiu, Chenchen Li, Junfeng Zhang.

**Formal analysis:** Li Qiu, Junfeng Zhang.

**Funding acquisition:** Yanli Wang.

**Investigation:** Li Qiu, Chenchen Li, Wen He, Xuelian Yin, Lin Zhan, Junfeng Zhang.

**Methodology:** Li Qiu, Chenchen Li, Junfeng Zhang.

**Supervision:** Yanli Wang.

**Writing – original draft:** Li Qiu, Chenchen Li, Junfeng Zhang.

**Writing – review & editing:** Li Qiu.

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
