## [Decision Letter · Decision Letter 0]

12 Oct 2022

PONE-D-22-24378Changes in diet, exercise and psychology of the quarantined population during the COVID-19 outbreak in ShanghaiPLOS ONE

Dear Dr. Wang,

Thank you for submitting your manuscript to PLOS ONE. After careful consideration, we feel that it has merit but does not fully meet PLOS ONE’s publication criteria as it currently stands. Therefore, we invite you to submit a revised version of the manuscript that addresses the points raised during the review process.

We look forward to receiving your revised manuscript.

Kind regards,

Bader A. Alqahtani

Academic Editor

PLOS ONE

Journal Requirements:

" ext-link-type="uri" xlink:type="simple">https://journals.plos.org/plosone/s/file?id=ba62/PLOSOne_formatting_sample_title_authors_affiliations.pdf"

Reviewers' comments:

Reviewer's Responses to Questions

**Comments to the Author**

1. Is the manuscript technically sound, and do the data support the conclusions?

Reviewer #1: Partly

Reviewer #2: Partly

2. Has the statistical analysis been performed appropriately and rigorously? 

Reviewer #1: No

Reviewer #2: Yes

3. Have the authors made all data underlying the findings in their manuscript fully available?

Reviewer #1: Yes

Reviewer #2: Yes

4. Is the manuscript presented in an intelligible fashion and written in standard English?

Reviewer #1: Yes

Reviewer #2: Yes

5. Review Comments to the Author

Reviewer #1: This study reports on changes in the diet, physical activity levels, and psychological status of the quarantined population during the COVID-19 outbreak in Shanghai, China. While the premise of this study is relevant and looks important, there are some comments/suggestions that the authors need to address before re-submission.

1. The background statement in the abstract should summarize the latest information on the topic; key phrases that pique interest, which lead to a clear definition of the study objectives. This is currently lacking.

2. I see that the authors have compromised on the results section of the abstract, which may be due to the word limits. As you have several significant findings, please, make sure you included as many as you possibly can, and consider mentioning statistical indicators.

3. There is little baseline knowledge on dietary, physical activity, and psychological status changes in the introduction. Authors need to develop further context and provide more information in the context of the previous studies and explain why their research is important. You may find the following studies helpful.

- Jia P, Liu L, Xie X, Yuan C, Chen H, Guo B, Zhou J, Yang S. Changes in dietary patterns among youths in China during COVID-19 epidemic: The COVID-19 impact on lifestyle change survey (COINLICS). Appetite. 2021 Mar 1;158:105015. doi: 10.1016/j.appet.2020.105015.

- Elnaggar RK, Alqahtani BA, Mahmoud WS, Elfakharany MS. Prospective analysis of physical activity levels and associated fitness factors amid COVID-19 pandemic and social-distancing rules. A special focus on adolescents. Sci Sports. 2022 Apr;37(2):131-138. doi: 10.1016/j.scispo.2021.07.002.

- Yu B, Zhang D, Yu W, Luo M, Yang S, Jia P. Impacts of lockdown on dietary patterns among youths in China: the COVID-19 Impact on Lifestyle Change Survey. Public Health Nutr. 2021 Aug;24(11):3221-3232. doi: 10.1017/S1368980020005170.

- Elnaggar RK, Alqahtani BA, Mahmoud WS, Elfakharany MS. Physical Activity in Adolescents During the Social Distancing Policies of the COVID-19 Pandemic. Asia Pac J Public Health. 2020 Nov;32(8):491-494. doi: 10.1177/1010539520963564.

- Yang GY, Lin XL, Fang AP, Zhu HL. Eating Habits and Lifestyles during the Initial Stage of the COVID-19 Lockdown in China: A Cross-Sectional Study. Nutrients. 2021 Mar 17;13(3):970. doi: 10.3390/nu13030970.

- Hossain MM, Tasnim S, Sultana A, Faizah F, Mazumder H, Zou L, McKyer ELJ, Ahmed HU, Ma P. Epidemiology of mental health problems in COVID-19: a review. F1000Res. 2020 Jun 23;9:636. doi: 10.12688/f1000research.24457.1.

- Dong L, Bouey J. Public Mental Health Crisis during COVID-19 Pandemic, China. Emerg Infect Dis. 2020 Jul;26(7):1616-1618. doi: 10.3201/eid2607.200407.

4. The 1st paragraph in the introduction section is a bit long. Please revise for brevity, directness, and conciseness.

5. Once again, the introduction should end with a straightforward and clear statement of study objectives. This is crucial to help readers make proper inferences and follow other parts of the study based on the study objectives.

6. I would have liked it if the authors had merely offered a summary of the survey's questions in the Methods under the “Measures” subheading.

7. Although 625 respondents appear to be reasonable, I am not sure if this study is powered enough for the study design and outcome measures.

8. You looked at how the respondent changed their behaviors from before to after closed management. Does the independent t-test fit this analysis?

9. I am not clear how Figure 3 illustrates the difference in exercise mode before and after closed management.

10. Discussion - The authors should discuss in-depth, focusing on the interpretation of their results in the context of the previous evidence and the implication of the study findings.

11. How can health care authorities in Shanghai make use of the results presented herein to facilitate access to care and for policy-making? Consider commenting on this point in the “Discussion” section.

12. Motivate on the study's merits, limitations, and how cautious readers should be while interpreting the results and give suggestions for future work.

Reviewer #2: This study aims to understand the changes in diet, exercise, and psychology of the population. The topic is interesting and attractive. However, the manuscript contains major issues that should be addressed:

Abstract:

- This section lacks its main structures (Background, objectives, methods, results, and conclusion).

- Add clear objectives.

- Add the study design.

- Add detailed information about the participants.

- Add the main significant findings, including p-values.

- Add a precise conclusion and future directions.

Introduction:

- Explain the rationale of the study. Please delete information unrelated to the objectives. Kindly focus on three elements of the introduction.

a. What is known about the topic? (Background)

b. What is not known? (The research problem)

c. Why the study was done? (Justification)

- Objective is not clear as mentioned above.

- Add the hypothesis of the study.

Methods:

- The study design, ethics, and setting are not clear.

- How and who administrates the data collection?

- How did you achieve the validity and reliability of the outcome measures?

- For statistical analysis, explain all methods used in detail and add the software used.

- Please, re-frame the components (SPICES) for methods

a. Study design, setting, sample size

b. Participants

c. Issue of interest (exposure)

d. Comparison

e. Ethics and endpoint

f. Statistical analysis

- What were the eligibility criteria for participants?

- Mention the settings and locations where the data were collected.

- How was the sample size determined?

- Who enrolled participants?

- Who assigned participants?

6. PLOS authors have the option to publish the peer review history of their article (what does this mean?). If published, this will include your full peer review and any attached files.

Reviewer #1: No

Reviewer #2: **Yes: **Walid Kamal Abdelbasset

---

## [Author Response · Author response to Decision Letter 0]

25 Nov 2022

Dear Editor,

Thank you for your letter which enclosed the reviewer’s comments concerning our manuscript entitled “Changes in diet, exercise and psychology of the quarantined population during the COVID-19 outbreak in Shanghai” (Research Article, PONE-D-22-24378). Those comments are all valuable and very helpful for revising and improving our paper, as well as the important guiding significance to our researches. We have studied comments carefully and have made correction which we hope meet with approval. The responds to the reviewer’s comments are as follows:

Reviewer #1: This study reports on changes in the diet, physical activity levels, and psychological status of the quarantined population during the COVID-19 outbreak in Shanghai, China. While the premise of this study is relevant and looks important, there are some comments/suggestions that the authors need to address before resubmission.

1. Comment: The background statement in the abstract should summarize the latest information on the topic; key phrases that pique interest, which lead to a clear definition of the study objectives. This is currently lacking. 

Response:

We gratefully appreciate for your comment. In the background statement section of the abstract, we reedited the language, summarized the research background of this paper, reflected the important information of the research background, and stimulated readers' interest through some data. (The contents of the correction are highlighted in red in the abstract on page 1 of the manuscript.)

2. Comment: I see that the authors have compromised on the results section of the abstract, which may be due to the word limits. As you have several significant findings, please, make sure you included as many as you possibly can, and consider mentioning statistical indicators.

Response:

We gratefully appreciate for your comment. We supplement the results in the abstract in more detail. Our research results will be summarized and summarized in more detail, and some figures will be used to clearly show our findings. (The contents of the correction are highlighted in red in the abstract on page 1 of the manuscript.)

3. Comment: There is little baseline knowledge on dietary, physical activity, and psychological status changes in the introduction. Authors need to develop further context and provide more information in the context of the previous studies and explain why their research is important. You may find the following studies helpful.

Response:

We gratefully appreciate for your comment. Thank you very much for providing these references. We have carefully read these articles and believe that their findings and results are of great help to our research and provide us with a lot of important information. In the introduction of this paper, we add the significance of the research findings of these articles to this paper and explain the importance of the research findings of these articles. (The contents of the correction are highlighted in red in the Introduction on page 2-3 and References on page 14-17 of the manuscript.)

- Jia P, Liu L, Xie X, Yuan C, Chen H, Guo B, Zhou J, Yang S. Changes in dietary patterns among youths in China during COVID-19 epidemic: The COVID-19 impact on lifestyle change survey (COINLICS). Appetite. 2021 Mar 1;158:105015. doi: 10.1016/j.appet.2020.105015.

- Elnaggar RK, Alqahtani BA, Mahmoud WS, Elfakharany MS. Prospective analysis of physical activity levels and associated fitness factors amid COVID-19 pandemic and social-distancing rules. A special focus on adolescents. Sci Sports. 2022 Apr;37(2):131-138. doi: 10.1016/j.scispo.2021.07.002.

- Yu B, Zhang D, Yu W, Luo M, Yang S, Jia P. Impacts of lockdown on dietary patterns among youths in China: the COVID-19 Impact on Lifestyle Change Survey. Public Health Nutr. 2021 Aug;24(11):3221-3232. doi: 10.1017/S1368980020005170.

- Elnaggar RK, Alqahtani BA, Mahmoud WS, Elfakharany MS. Physical Activity in Adolescents During the Social Distancing Policies of the COVID-19 Pandemic. Asia Pac J Public Health. 2020 Nov;32(8):491-494. doi: 10.1177/1010539520963564.

- Yang GY, Lin XL, Fang AP, Zhu HL. Eating Habits and Lifestyles during the Initial Stage of the COVID-19 Lockdown in China: A Cross-Sectional Study. Nutrients. 2021 Mar 17;13(3):970. doi: 10.3390/nu13030970.

- Hossain MM, Tasnim S, Sultana A, Faizah F, Mazumder H, Zou L, McKyer ELJ, Ahmed HU, Ma P. Epidemiology of mental health problems in COVID-19: a review. F1000Res. 2020 Jun 23;9:636. doi: 10.12688/f1000research.24457.1.

- Dong L, Bouey J. Public Mental Health Crisis during COVID-19 Pandemic, China. Emerg Infect Dis. 2020 Jul;26(7):1616-1618. doi: 10.3201/eid2607.200407.

4. Comment: The 1st paragraph in the introduction section is a bit long. Please revise for brevity, directness, and conciseness.

Response:

We gratefully appreciate for your comment. We cut the background information in the first paragraph of the introduction that was not concise enough, so that the research background in the introduction became more concise, direct and concise. (The contents of the correction are highlighted in red in the Introduction on page 2-3 of the manuscript.)

5. Comment: Once again, the introduction should end with a straightforward and clear statement of study objectives. This is crucial to help readers make proper inferences and follow other parts of the study based on the study objectives.

Response:

We gratefully appreciate for your comment. We have modified the content of the third paragraph of the introduction and added a more direct and clear research purpose in this paragraph, hoping that such a statement will help readers understand the research purpose of this paper. (The contents of the correction are highlighted in red in the Introduction on page 2-3 of the manuscript.)

6. Comment: I would have liked it if the authors had merely offered a summary of the survey's questions in the Methods under the “Measures” subheading. 

Response:

We gratefully appreciate for your comment. We added 10 main questions from the questionnaire used in the study in section 2.2 Measures. (The content added is shown in Table 1 below.)

Table 1. The main questions of the survey

Q: How many kinds of food do you eat every day?

A: 1.0-2 species 2.3-5 species 3.6-8 species 4.9-11 species 5.12 species or more

Q: How much vegetables do you eat every day?

A: 1. 0-50 grams 2.51-100 grams 3.101-200 grams 4.201-299 grams 5.300 grams or more

Q: How much fruit do you eat every day?

A: 1. 0-50 grams 2. 51-100 grams 3. 101-150 grams 4. 151-199 grams 5. 200 grams or more

Q: How many hours do you exercise every week?

A: 1. Less than 1 hour 2. 1-2 hours 3. 2-3 hours 4. 3-4 hours 5. More than 4 hours

Q: What are your main sports activities?

A: 1. Track and field walking 2. Ball 3. Swimming 4. Fitness equipment and gym 5. Other

Q: How many steps do you take every day?

A: 1.0-1500 steps 2.1501-3000 steps 3.3001-4500 steps 4.4501-5999 steps 5.6000 steps and above

Q: Little interest or pleasure in doing things?

A: 1. Strongly disagree 2. Disagree 3. Neither agree nor disagree 4. Agree 5. Strongly agree

Q: Feeling down, depressed, or hopeless?

A: 1. Strongly disagree 2. Disagree 3. Neither agree nor disagree 4. Agree 5. Strongly agree

Q: Trouble falling or staying asleep, or sleeping too much?

A: 1. Strongly disagree 2. Disagree 3. Neither agree nor disagree 4. Agree 5. Strongly agree

Q: Feeling tired or having little energy?

A: 1. Strongly disagree 2. Disagree 3. Neither agree nor disagree 4. Agree 5. Strongly agree

7. Comment: Although 625 respondents appear to be reasonable, I am not sure if this study is powered enough for the study design and outcome measures. 

Response:

We gratefully appreciate for your comment. A total of 671 questionnaires were collected, of which 625 were valid, with an effective recovery rate of 93.14%. Our survey group was concentrated in the population under quarantine control, and we distributed the questionnaire to this group based on this situation. Before conducting this study, we reviewed some literature materials and found that relevant articles in this field also conducted similar surveys. For example, a study on Wuhan collected 376 questionnaires[1], of which 339 were valid (90.2%); In a Japanese study[2], researchers looked at 244 and 220 people; Another study looked at the eating habits and psychological state of 176 Italian college students during home isolation[3].

Their research also has quite important findings, therefore, the number of our questionnaires should be able to support this research.

References

1. He M, Xian Y, Lv XD et al. Changes in Body Weight, Physical Activity, and Lifestyle During the Semi-lockdown Period After the Outbreak of COVID-19 in China: An Online Survey. Disaster Medicine and Public Health Preparedness. 2021, 15(2): E23-E28.

2.Okuyama J, Seto S, Fukuda Y et al. Life Alterations and Stress During the COVID-19 Pandemic in Japan: Two-Time Comparison. Journal of Disaster Research. 2022, 17(1): 43-50.

3. Amatori S, Zeppa SD, Preti A et al. Dietary Habits and Psychological States during COVID-19 Home Isolation in Italian College Students: The Role of Physical Exercise[J]. Nutrients. 2020, 12(12).

8. Comment: You looked at how the respondent changed their behaviors from before to after closed management. Does the independent t-test fit this analysis?

Response:

We gratefully appreciate for your comment. During the analysis of this study, we conducted an independent t-test on the samples, and the Results showed that there were significant differences in the behavior changes of the respondents before and after the closed management (as shown in the following Table 2. Table 3. Table 4.). (We also added this result in each subchapter of 3 Results in the main text.)

Table 2. Differences in diet before and after closed management

 Before the closed management After the closed management p-value T-test

Sig. (two-sided test)

 mean min max mean min max 

Balanced match of meat and vegetables, intake of various kinds of foods 3.90 1 5 3.67 1 5 0.000 0.000

The types of food you eat each day 3.50 1 5 3.24 1 5 0.002 0.000

You usually eat how full 3.78 1 5 3.64 1 5 0.001 0.019

Your food tastes light 3.50 1 5 3.64 1 5 0.023 0.038

You eat salt every day 1.39 1 2 1.36 1 2 0.037 0.000

You eat sugar every day 1.83 1 3 1.79 1 3 0.000 0.000

You often eat fruits and vegetables 3.84 1 5 3.45 1 5 0.000 0.000

You eat vegetables every day 3.23 1 5 2.94 1 5 0.000 0.000

Table 3. Differences in exercise before and after closed management

 Before the closed management After the closed management p-value T-test

Sig. (two-sided test)

 mean min max mean min max 

Total exercise time per week 3.19 1 5 2.27 1 5 0.000 0.000

The number of steps taken each day 3.60 1 5 2.25 1 5 0.012 0.000

Table 4. Psychological changes before and after closed management

 Before the closed management After the closed management p-value T-test

Sig.(two-sided test)

 mean min max mean min max 

Often feeling down, depressed or hopeless 2.78 1 5 2.97 1 5 0.000 0.006

Often feel tired or low in energy 2.73 1 5 3.10 1 5 0.002 0.000

Mood swings are high and easy to rise and fall 2.79 1 5 2.90 1 5 0.000 0.004

It's hard to control your unhappiness 2.77 1 5 2.82 1 5 0.008 0.000

Always feeling passive and unmotivated 2.70 1 5 2.84 1 5 0.001 0.049

9. Comment: I am not clear how Figure 3 illustrates the difference in exercise mode before and after closed management. 

Response:

We gratefully appreciate for your comment. We are sorry that we did not clearly explain the significance of the difference of exercise style before and after the closure in Figure 3 in the first draft. Therefore, we made a detailed explanation of the exercise mode after the closure in 3.3 Changes in physical activity in Figure 3, as described below. However, after the lockdown management, people can hardly carry out swimming exercise; Track and walking decreased by 2 percent and ball games by 4 percent; Three percent increased their use of fitness equipment; Another 15 percent added other forms of exercise, such as calisthenics, that can be done indoors without equipment. (The contents of the correction are highlighted in red in the 3.3 Changes in physical activity on page 7-9 of the manuscript.)

Figure 3. Exercise mode before and after closed management.

10. Comment: Discussion - The authors should discuss in-depth, focusing on the interpretation of their results in the context of the previous evidence and the implication of the study findings.

Response:

We gratefully appreciate for your comment. Combined with past research findings and the research results of this paper, we make a more in-depth analysis and discussion in the discussion section. (The contents of the correction are highlighted in red in the Discussion on page 10-12 of the manuscript.)

11. Comment: How can health care authorities in Shanghai make use of the results presented herein to facilitate access to care and for policy-making? Consider commenting on this point in the “Discussion” section.

Response:

We gratefully appreciate for your comment. In the "Discussion" part, we add some practical suggestions, which are put forward based on the research findings and results of this paper. We hope that these suggestions can provide some references for relevant departments to formulate relevant policies in the future, as described below. For example: communities can provide services such as delivering fresh vegetables and fruits to people under lockdown. For the closed management of the population can be implemented by time partition measures to provide people with outdoor exercise opportunities; During the lockdown period, psychological counseling should be strengthened for relevant personnel, and psychological hotlines should be kept unblocked or online psychological counseling channels should be opened. (The contents of the correction are highlighted in red in the Discussion on page 10-12 of the manuscript.)

12. Comment: Motivate on the study's merits, limitations, and how cautious readers should be while interpreting the results and give suggestions for future work.

Response:

We gratefully appreciate for your comment. At the end of the text, we add some advantages and innovations of this research, and also carefully elaborate the limitations of this research. For readers, we also offer some suggestions for a cautious interpretation of the findings. Finally, we also provide some useful suggestions for the future related research work, hoping to contribute to the future development of this field, as described below. The strength of this study is that it discusses the changes in diet, exercise and psychology brought about by closed management. However, this study also has some limitations, such as a short research period, and due to closed management measures, the questionnaire can only be distributed online, so some people who do not often use social media platforms are not included in the survey. In addition, this study mainly explored the changes in diet, exercise and psychological state before and after lockdown, but did not involve the causes of these changes. The influence of closed management on people is not only in the three aspects of diet, exercise and psychological state, but also in the aspects of people's work, study and happiness. This will be questions that researchers can continue to explore in the future. (The contents of the correction are highlighted in red in the Discussion on page 10-12 of the manuscript.)

Reviewer #2: This study aims to understand the changes in diet, exercise, and psychology of the population. The topic is interesting and attractive. However, the manuscript contains major issues that should be addressed:

Abstract:

- This section lacks its main structures (Background, objectives, methods, results, and conclusion).

- Add clear objectives.

- Add the study design.

- Add detailed information about the participants.

- Add the main significant findings, including p-values.

- Add a precise conclusion and future directions.

Response:

We gratefully appreciate for your comment. We divide the abstract into five parts, which are background, objectives, methods, results, and conclusion. For the content of the abstract, we have made the following changes. Firstly, we clearly describe the research background, then clarify the research objectives, methods and participants' information, and also add the important findings of this study, P-value and related data, and add conclusions and recommendations at the end of the abstract. (The contents of the correction are highlighted in red in the abstract on page 1 of the manuscript.)

Introduction:

- Explain the rationale of the study. Please delete information unrelated to the objectives. Kindly focus on three elements of the introduction.

a. What is known about the topic? (Background)

b. What is not known? (The research problem)

c. Why the study was done? (Justification)

- Objective is not clear as mentioned above.

- Add the hypothesis of the study.

Response:

We gratefully appreciate for your comment. We deleted the tedious background information in the introduction and made the research background concise and clear. Secondly, the research objectives and problems to be solved in this paper are clarified. We also added our reasons for doing the study. (The contents of the correction are highlighted in red in the Introduction on page 2-3 of the manuscript.)

Methods:

- The study design, ethics, and setting are not clear.

- How and who administrates the data collection?

- How did you achieve the validity and reliability of the outcome measures?

- For statistical analysis, explain all methods used in detail and add the software used.

- Please, re-frame the components (SPICES) for methods

a. Study design, setting, sample size

b. Participants

c. Issue of interest (exposure)

d. Comparison

e. Ethics and endpoint

f. Statistical analysis

- What were the eligibility criteria for participants?

- Mention the settings and locations where the data were collected.

- How was the sample size determined?

- Who enrolled participants?

- Who assigned participants?

Response:

We gratefully appreciate for your comment. The research protocol was approved by the ethics committee of Shanghai University. All participants provide informed consent prior to taking part in this study. The eligibility criteria for participants are people in universities or Communities in Shanghai who have experienced lockdown or quarantine due to the studio. According to the division of labor among the members, Qiu Li and Xu Yajun mainly manage the data collection. Statistical data were analyzed using SPSS. A total of 671 questionnaires were collected, of which 625 were valid, with an effective recovery rate of 93.14%. Our respondents were concentrated in the population under quarantine control, and we distributed the questionnaire in this group based on this situation, so as to ensure the validity and reliability of the outcome. As for the sample size, we found that 376 questionnaires were collected in a study on Wuhan, among which 339 were valid (90.2%). In a Japanese study, researchers looked at 244 and 220 people; Another study looked at the eating habits and psychological state of 176 Italian college students during home isolation. Their study also had quite important findings, so our questionnaire number of 625 can support this study. (The contents of the correction are highlighted in red in the 2 Materials and Methods on page 3-5 of the manuscript.)

Thank you again for your positive and constructive comments and suggestions on our manuscript. We hope you will find our revised manuscript acceptable for publication.

NOTE: All modifications to the manuscript are marked with red in the Word.

---

## [Decision Letter · Decision Letter 1]

28 Mar 2023

PONE-D-22-24378R1Changes in diet, exercise and psychology of the quarantined population during the COVID-19 outbreak in ShanghaiPLOS ONE

Dear Dr. Wang,

Thank you for submitting your manuscript to PLOS ONE. After careful consideration, we feel that it has merit but does not fully meet PLOS ONE’s publication criteria as it currently stands. Therefore, we invite you to submit a revised version of the manuscript that addresses the points raised during the review process.

We look forward to receiving your revised manuscript.

Kind regards,

Bader A. Alqahtani

Academic Editor

PLOS ONE

Journal Requirements:

Reviewers' comments:

Reviewer's Responses to Questions

**Comments to the Author**

1. If the authors have adequately addressed your comments raised in a previous round of review and you feel that this manuscript is now acceptable for publication, you may indicate that here to bypass the “Comments to the Author” section, enter your conflict of interest statement in the “Confidential to Editor” section, and submit your "Accept" recommendation.

Reviewer #1: All comments have been addressed

Reviewer #3: (No Response)

2. Is the manuscript technically sound, and do the data support the conclusions?

Reviewer #1: Yes

Reviewer #3: Yes

3. Has the statistical analysis been performed appropriately and rigorously? 

Reviewer #1: Yes

Reviewer #3: Yes

4. Have the authors made all data underlying the findings in their manuscript fully available?

Reviewer #1: Yes

Reviewer #3: Yes

5. Is the manuscript presented in an intelligible fashion and written in standard English?

Reviewer #1: Yes

Reviewer #3: Yes

6. Review Comments to the Author

Reviewer #1: The authors satisfactorily addressed the issues brought up in the previous round.

The manuscript significantly improved, and in my opinion, it is publishable in the present format.

Reviewer #3: This is my first time to review this paper. I have a few comments.

1) Under the Introduction, please mention how lockdown affect other countries besides China:

Search PubMed for: Partial Lockdown and Factors that were associated with an increased level of depression, stress, and anxiety were being single, separated, or widowed, a higher education level, a larger family size, loss of jobs and being in contact with potential COVID-19 patients

2) Under the discussion, please state limitation of this study as it did not measure COVID-19 burnout and discuss the following finding:

Search PubMed for: Burnout is an important public health issue at times of the COVID-19 pandemic. Current measures which focus on work-based burnout have limitations in length and/or relevance. When stepping into the post-pandemic as a new Norm Era, the burnout scale for the general population is urgently needed to fill the gap. This study aimed to develop a COVID-19 Burnout Views Scale (COVID-19 BVS) to measure burnout views of the general public in a Chinese context and examine its psychometric properties.

3) The authors mentioned about psychological interventions and it must be online during the lockdown period. Please refer to the following:

The most evidence-based treatment is cognitive behaviour therapy (CBT), especially Internet CBT that can prevent the spread of infection during the pandemic.

Use of Cognitive Behavior Therapy (CBT) to treat psychiatric symptoms during COVID-19:

Mental Health Strategies to Combat the Psychological Impact of COVID-19 Beyond Paranoia and Panic. Ann Acad Med Singapore. 2020;49(3):155‐160.

Cost-effectiveness of iCBT:

Moodle: The cost effective solution for internet cognitive behavioral therapy (I-CBT) interventions. Technol Health Care. 2017;25(1):163-165. doi: 10.3233/THC-161261. PMID: 27689560.

Internet CBT can treat psychiatric symptoms such as insomnia:

Efficacy of digital cognitive behavioural therapy for insomnia: a meta-analysis of randomised controlled trials. Sleep Med. 2020 Aug 26;75:315-325. doi: 10.1016/j.sleep.2020.08.020. Epub ahead of print. PMID: 32950013.

7. PLOS authors have the option to publish the peer review history of their article (what does this mean?). If published, this will include your full peer review and any attached files.

Reviewer #1: No

Reviewer #3: No

---

## [Author Response · Author response to Decision Letter 1]

31 Mar 2023

Dear Editor and reviewers,

Thank you for your letter which enclosed the reviewer’s comments concerning our manuscript entitled “Changes in diet, exercise and psychology of the quarantined population during the COVID-19 outbreak in Shanghai” (Research Article, PONE-D-22-24378). Those comments are all valuable and very helpful for revising and improving our paper, as well as the important guiding significance to our researches. We have studied comments carefully and have made correction which we hope meet with approval. The responds to the reviewer’s comments are as follows:

Reviewer #3: This is my first time to review this paper. I have a few comments.

1. Comment: Under the Introduction, please mention how lockdown affect other countries besides China:

Search PubMed for: Partial Lockdown and Factors that were associated with an increased level of depression, stress, and anxiety were being single, separated, or widowed, a higher education level, a larger family size, loss of jobs and being in contact with potential COVID-19 patients

Response:

We gratefully appreciate for your comment. We searched for the information you mentioned. In the introduction section, we add content about the impact of the lockdown on countries other than China. 

The contents of the correction is as followed: A study in Vietnam found significantly higher levels of depression, anxiety and stress among people in lockdown due to being single, separated or widowed, higher levels of education, larger family size, unemployment, and contact with potential COVID-19 patients [19]. A study from France says the lockdown affects people's mood, with hope and anxiety being the two ways to deal with uncertainty [20]. 

References:

19.Le HT, Lai AJX, Sun J et al. Anxiety and Depression Among People Under the Nationwide Partial Lockdown in Vietnam[J]. Frontiers in public health. 2020, 8: 589359.

20.Fouques D, Castro D, Mouret M et al. Perceptions of the Post First-Lockdown Era in the Current Covid-19 Pandemic: Quantitative and Qualitative Survey of the French Population. Frontiers in Psychology. 2021, 12. DOI: 10.3389/fpsyg.2021.668961 

(The contents of the correction are highlighted in red in the Introduction on page 2 and References on page 15 of the manuscript.)

2. Comment: Under the discussion, please state limitation of this study as it did not measure COVID-19 burnout and discuss the following finding:

Search PubMed for: Burnout is an important public health issue at times of the COVID-19 pandemic. Current measures which focus on work-based burnout have limitations in length and/or relevance. When stepping into the post-pandemic as a new Norm Era, the burnout scale for the general population is urgently needed to fill the gap. This study aimed to develop a COVID-19 Burnout Views Scale (COVID-19 BVS) to measure burnout views of the general public in a Chinese context and examine its psychometric properties.

Response:

We gratefully appreciate for your comment. In the "Discussion" part, we add some limitation of our study as it did not measure COVID-19 burnout. We searched and learned about the burnout you mentioned. These are also discussed in the "Discussion" section. 

The contents of the correction is as followed: In the part of investigating the psychological state of the population under lockdown, this study did not involve the investigation of burnout in the population. Burnout is an important public health problem, and it is very important to study the burnout of the general public when entering the post-epidemic era.

References:

46.Lau SSS, Ho CCY, Pang RCK et al. Measurement of burnout during the prolonged pandemic in the Chinese zero-COVID context: COVID-19 burnout views scale[J]. Frontiers in Public Health. 2022, 10.

(The contents of the correction are highlighted in red in the Discussion on page 12 and References on page 17 of the manuscript.)

3. Comment: The authors mentioned about psychological interventions and it must be online during the lockdown period. Please refer to the following:

The most evidence-based treatment is cognitive behaviour therapy (CBT), especially Internet CBT that can prevent the spread of infection during the pandemic.

Use of Cognitive Behavior Therapy (CBT) to treat psychiatric symptoms during COVID-19:

Mental Health Strategies to Combat the Psychological Impact of COVID-19 Beyond Paranoia and Panic. Ann Acad Med Singapore. 2020;49(3):155‐160.

Cost-effectiveness of iCBT:

Moodle: The cost effective solution for internet cognitive behavioral therapy (I-CBT) interventions. Technol Health Care. 2017;25(1):163-165. doi: 10.3233/THC-161261. PMID: 27689560.

Internet CBT can treat psychiatric symptoms such as insomnia:

Efficacy of digital cognitive behavioural therapy for insomnia: a meta-analysis of randomised controlled trials. Sleep Med. 2020 Aug 26;75:315-325. doi: 10.1016/j.sleep.2020.08.020. Epub ahead of print. PMID: 32950013.

Response:

We gratefully appreciate for your comment. Thank you very much for providing these references. We have carefully read these articles and believe that their findings and results are of great help to our research and provide us with a lot of important information. In the introduction and discussion of this paper, we add the significance of the research findings of these articles to this paper and explain the importance of the research findings of these articles. 

References:

18. Ho CSH, Chee CYI, Ho RCM. Mental Health Strategies to Combat the Psychological Impact of Coronavirus Disease 2019 (COVID-19) Beyond Paranoia and Panic[J]. Annals Academy of Medicine Singapore. 2020, 49(3): 155-160.

29. Zhang MWB, Ho RCM. Moodle: The cost effective solution for internet cognitive behavioral therapy (I-CBT) interventions[J]. Technology and Health Care. 2017, 25(1): 163-165.

45.Soh HL, Ho RC, Ho CS et al. Efficacy of digital cognitive behavioural therapy for insomnia: a meta-analysis of randomised controlled trials[J]. Sleep Medicine. 2020, 75: 315-325.

 (The contents of the correction are highlighted in red in the Introduction on page 2-3, Discussion on page 11-12 and References on page 15-17 of the manuscript.)

Thank you again for your positive and constructive comments and suggestions on our manuscript. We hope you will find our revised manuscript acceptable for publication.

NOTE: All modifications to the manuscript are marked with red in the Word.

---

## [Decision Letter · Decision Letter 2]

10 Apr 2023

Changes in diet, exercise and psychology of the quarantined population during the COVID-19 outbreak in Shanghai

PONE-D-22-24378R2

Dear Dr. Wang,

We’re pleased to inform you that your manuscript has been judged scientifically suitable for publication and will be formally accepted for publication once it meets all outstanding technical requirements.

Kind regards,

Bader A. Alqahtani

Academic Editor

PLOS ONE

Additional Editor Comments (optional):

Reviewers' comments:

Reviewer's Responses to Questions

**Comments to the Author**

1. If the authors have adequately addressed your comments raised in a previous round of review and you feel that this manuscript is now acceptable for publication, you may indicate that here to bypass the “Comments to the Author” section, enter your conflict of interest statement in the “Confidential to Editor” section, and submit your "Accept" recommendation.

Reviewer #3: All comments have been addressed

2. Is the manuscript technically sound, and do the data support the conclusions?

Reviewer #3: Yes

3. Has the statistical analysis been performed appropriately and rigorously? 

Reviewer #3: Yes

4. Have the authors made all data underlying the findings in their manuscript fully available?

Reviewer #3: Yes

5. Is the manuscript presented in an intelligible fashion and written in standard English?

Reviewer #3: Yes

6. Review Comments to the Author

Reviewer #3: I recommend publication for the paper "Changes in diet, exercise and psychology of the quarantined population during the COVID-19 outbreak in Shanghai"

7. PLOS authors have the option to publish the peer review history of their article (what does this mean?). If published, this will include your full peer review and any attached files.

Reviewer #3: No

---

## [Editor Report · Acceptance letter]

24 Jul 2023

PONE-D-22-24378R2 

Changes in diet, exercise and psychology of the quarantined population during the COVID-19 outbreak in Shanghai 

Dear Dr. Wang:

I'm pleased to inform you that your manuscript has been deemed suitable for publication in PLOS ONE. Congratulations! Your manuscript is now with our production department. 

Kind regards, 

on behalf of

Dr. Bader A. Alqahtani 

Academic Editor

PLOS ONE